# Baloxavir treatment of ferrets infected with influenza A(H1N1)pdm09 virus reduces onward transmission

**Leo Yi Yang Lee**[1⦿], **Jie Zhou**[2⦿], **Rebecca Frise**[2], **Daniel H. Goldhill**[2],
**Paulina Koszalka**[1,3], **Edin J. Mifsud**[1,4], **Kaoru Baba**[5], **Takahiro Noda**[5], **Yoshinori Ando**[6],
**Kenji Sato**[6], **Aoe-Ishikawa Yuki**[5], **Takao Shishido**[6], **Takeki Uehara**[6], **Steffen Wildum**[7],
**Elke Zwanziger**[7], **Neil Collinson**[8], **Klaus Kuhlbusch**[7], **Barry Clinch**[8], **Aeron C. Hurt**[1,4,7‡]*,
**Wendy S. Barclay**[2‡]*

**1** WHO Collaborating Centre for Reference and Research on Influenza, at the Peter Doherty Institute for Infection and Immunity, Melbourne, Australia, **2** Department of Infectious Disease, Imperial College London, London, United Kingdom, **3** Biomedicine Discovery Institute & Department of Microbiology, Monash University, Victoria 3800, Australia, **4** Department of Microbiology and Immunology, University of Melbourne, at the Peter Doherty Institute for Infection and Immunity, Parkville, Australia, **5** Shionogi TechnoAdvance Research, Co., Ltd., Osaka, Japan, **6** Shionogi & Co., Ltd, Osaka, Japan, **7** F. Hoffmann-La Roche Ltd, Basel, Switzerland, **8** Roche Products Ltd, Welwyn, United Kingdom

⦿ These authors contributed equally to this work.
‡ These authors are joint senior authors on this work.
* aeron.hurt@roche.com (ACH); w.barclay@imperial.ac.uk (WSB)

**Data Availability Statement:** All relevant data are within the manuscript and its Supporting Information files.

## Abstract

Influenza viruses cause seasonal outbreaks and pose a continuous pandemic threat. Although vaccines are available for influenza control, their efficacy varies each season and a vaccine for a novel pandemic virus manufactured using current technology will not be available fast enough to mitigate the effect of the first pandemic wave. Antivirals can be effective against many different influenza viruses but have not thus far been used extensively for outbreak control. Baloxavir, a recently licensed antiviral drug that targets the influenza virus endonuclease, has been shown to reduce virus shedding more effectively than oseltamivir, a widely used neuraminidase inhibitor drug. Thus it is possible that treatment with baloxavir might also interrupt onward virus transmission. To test this, we utilized the ferret model, which is the most commonly used animal model to study influenza virus transmission. We established a subcutaneous baloxavir administration method in ferrets which achieved similar pharmacokinetics to the approved human oral dose. Transmission studies were then conducted in two different locations with different experimental setups to compare the onward transmission of A(H1N1)pdm09 virus from infected ferrets treated with baloxavir, oseltamivir or placebo to naïve sentinel ferrets exposed either indirectly in adjacent cages or directly by co-housing. We found that baloxavir treatment reduced infectious viral shedding in the upper respiratory tract of ferrets compared to placebo, and reduced the frequency of transmission amongst sentinels in both experimental setups, even when treatment was delayed until 2 days post-infection. In contrast, oseltamivir treatment did not substantially affect viral shedding or transmission compared to placebo. We did not detect the emergence of baloxavir-resistant variants in treated animals or in untreated sentinels.

**Funding:** The Melbourne WHO Collaborating Centre for Reference and Research on Influenza is supported by the Australian Government Department of Health. This work was funded by F. Hoffmann-La Roche (https://www.roche.com/) and Shionogi & Co. Ltd (https://www.shionogi.com/) who both played a role in the study design, data collection and analysis, decision to publish and preparation of the manuscript.

**Competing interests:** I have read the journal's policy and the authors of this manuscript have the following competing interests: Leo Yi Yang Lee, Jie Zhou, Rebecca Frise, Daniel H. Goldhill, Paulina Koszalka and Edin J. Mifsud have nothing to declare. Kaoru Baba, Takahiro Noda, Yoshinori Ando, Kenji Sato, Aoe-Ishikawa Yuki, Takao Shishido and Takeki Uehara are employees of Shionogi & Co. Ltd. Steffen Wildum, Elke Zwanziger, Neil Collinson, Klaus Kuhlbusch, Barry Clinch and Aeron C. Hurt are employees of F. Hoffmann La Roche Ltd. Wendy S. Barclay has received honoraria from Roche, Sanofi Pasteur and Seqirus.

Our results support the concept that antivirals which decrease viral shedding could also reduce influenza transmission in the community.

## Author summary

During seasonal influenza outbreaks and global pandemics, influenza can cause significant morbidity and mortality and spread rapidly. Influenza viruses constantly change, and the effectiveness of vaccination can be low if the match between the vaccine and circulating viruses is poor. However, antiviral drugs target conserved parts of the virus and therefore typically remain effective against new seasonal or pandemic strains of influenza. The new antiviral baloxavir is more effective than existing drugs, such as oseltamivir, in reducing the amount of virus particles produced by infected people, suggesting it might reduce the onward spread of influenza viruses to others. To test this, we developed an effective way to deliver baloxavir to ferrets, the best available animal model of influenza virus transmission. We then treated influenza-infected ferrets with baloxavir to determine if they were less likely to pass their virus onto healthy ferrets housed in the same cage, or in the adjacent cage. In both cases, we found that compared to oseltamivir or placebo treatment, infected ferrets treated with baloxavir produced fewer virus particles and were less likely to transmit virus to healthy ferrets. Our results suggest baloxavir can contribute to the early control of influenza outbreaks by limiting community-based viral spread.

## Introduction

Seasonal influenza is a major cause of morbidity and mortality, resulting in 3–5 million cases of severe respiratory disease and up to 650,000 deaths worldwide each year [1]. Vaccination is a major component of seasonal influenza management, but vaccine effectiveness varies each season; the current range is 33% to 67% [2], and can be lower if the predominant circulating influenza A virus subtype is H3N2. In many countries, uptake of the vaccine remains low. A future pandemic caused by a novel influenza virus has the potential to cause tens of millions of deaths [3], but it is likely that a specific vaccine to a new pandemic virus would take at least six months to manufacture and distribute [4]. Antivirals therefore play an important role in the prevention and treatment of both seasonal and pandemic influenza.

The neuraminidase inhibitors (NAIs) are a class of influenza antivirals that includes the widely prescribed compound oseltamivir phosphate (OST), which was licensed in most countries in the world two decades ago. Baloxavir marboxil (BXM) is a new first-in-class single-dose influenza antiviral that inhibits the influenza cap-dependent endonuclease, preventing viral replication [5]. BXM is administered as an oral prodrug and is hydrolysed to the active metabolite, baloxavir acid (BXA), by arylacetamide deacetylase in the small intestine, blood and liver. BXM is currently licensed for use in Japan, the USA, and several other countries for the treatment of infections with influenza A or B viruses. In a phase 3 clinical trial in otherwise-healthy patients, a single dose of BXM reduced the time to alleviation of symptoms compared to placebo by 26 hours. Although this clinical benefit was comparable to that seen for OST, BXM-treated patients experienced a dramatic reduction in viral shedding that was not observed in either placebo- or OST-treated patients [6]. The median duration of infectious virus detection was only 24 hours following BXM treatment, compared with 72 and 96 hours following OST or placebo treatment, respectively. Households represent a critical setting for

the transmission of seasonal and pandemic influenza, as potentially susceptible contacts are in close proximity to primarily infected (index) patients shedding infectious virus [7]. There is limited evidence that treatment of index patients with NAIs may reduce the risk of secondary infection to household contacts without prophylaxis [8–11]. Therefore, we hypothesized that the faster reduction in viral shedding following baloxavir treatment would result in a greater reduction in the onward transmission of influenza viruses than OST.

Ferrets are considered the gold standard animal model for evaluating influenza infection and transmission as they are susceptible to human influenza strains, shed virus in the nasal cavity with similar kinetics to that seen in humans, show similar clinical signs to humans, and can readily transmit the virus to exposed contacts [12,13]. A detailed pharmacokinetic (PK) analysis of OST in ferrets has been previously completed and a dose established that aligns to the typical PK profile of OST observed in humans following standard dosing [14]. However this has not previously been completed for BXM/BXA in ferrets. Therefore, after establishing the pharmacokinetics of BXM/BXA in ferrets, we used the ferret model to evaluate the effect of BXA, OST or placebo treatment of ferrets infected with A(H1N1)pdm09 influenza viruses on onward transmission of virus to naïve sentinel ferrets.

## Materials and methods

The ferret transmission experiments were conducted in two independent laboratories: Imperial College London, London, UK, and The Peter Doherty Institute for Infection and Immunity, Melbourne, Australia, hereafter referred to as London and Melbourne, respectively. Where methods differed, details are provided under separate headings for the two laboratories.

### Compounds

BXM and BXA were synthesized and provided by Shionogi & Co. Ltd. BXM was suspended in water containing 5% (w/v) sodium dodecyl sulfate (SDS) and 10% (w/w) polysorbate 80 (Wako Pure Chemical Industries, Osaka, Japan). BXA was prepared in a 0.5 w/v % methyl cellulose (Sigma-Aldrich, Australia) solution using an agate mortar and pestle. Oseltamivir phosphate was purchased from Sequoia Research Products Ltd (Pangbourne, UK) and dissolved in a 1 g/mL sterile sugar solution to a concentration of 10 mg/mL.

### BXA pharmacokinetic analysis

Prior to conducting ferret infection/transmission experiments it was important to establish the most appropriate dose and route of administration of BXM or BXA in ferrets to achieve BXA levels equivalent to that seen in humans. For oral BXM delivery, female ferrets (*Mustela putorius furo*) (14–17 months of age) weighing 792–937 g (Japan SLC Inc.) were used. Suspension of BXM at 10 mg/5 mL/kg was delivered to fasted ferrets anesthetized by isoflurane using an intragastric tube to achieve a dose of 10 mg/kg. BXA was delivered in female ferrets (*Mustela putorius furo*) (28 months of age) weighing 650–802 g (Japan SLC Inc.) subcutaneously to four locations on the dorsal region (4 mg/kg per animal) using a 25-gauge needle and 1 mL syringe. To evaluate BXA concentrations, blood (approximately 60–100 μL) was collected into heparinized capillary tubes (Drummond Scientific Company, USA) and plasma obtained by centrifugation. The concentration of BXA in plasma was determined by LC-MS/MS (liquid chromatography with tandem mass spectrometry), consisting of a LC-20A system (Shimadzu Corporation, Japan) and an API 5000 (AB SCIEX, USA). Plasma samples were prepared by protein precipitation and chromatographic separation was performed on L-column 2 ODS metal free (3 μm, 2.0 mm i.d. × 50 mm, Chemicals Evaluation and Research Institute, Japan) at

40˚C. The binary mobile phases (0.1% formic acid in water and 0.1% formic acid in acetonitrile) were delivered at the total flow rate of 0.6 mL/min in gradient mode. The mass spectrometer was operated in electrospray ionization (ESI) positive polarity mode using multiple reaction monitoring (MRM). Precursor/product transitions (m/z) of 484/247 and 490/247 were monitored for BXA and internal standard, BXA-racemate-$d_4$ $^{18}O$, respectively. Calculations were based on peak-area ratios of BXA to internal standard. The analytical method was validated across the calibration range 0.5 to 500 ng/mL with respect to selectivity, recovery, accuracy, precision, and stability under a variety of conditions.

The concentrations of BXA in ferret plasma samples (collected using heparinized tubes) obtained from transmission studies were measured after protein precipitation using the same LC-MS/MS method as described above by Sumika Chemical Analysis Service, Ltd (Japan).

## Cells and viruses

**London experiments.** Madin-Darby Canine Kidney (MDCK) cells were grown in Dulbecco's modified Eagle's medium (DMEM) supplemented with 10% (v/v) fetal bovine serum and 1% penicillin/streptomycin (Sigma-Aldrich). Recombinant influenza A/England/195/2009 (H1N1)pdm09 virus was generated by reverse genetics as previously described [15].

**Melbourne experiments.** MDCK cells were grown in DMEM (Gibco) supplemented with 10% (v/v) fetal bovine serum (Bovogen Biologicals, Australia), 1x GlutaMAX (Gibco, USA), 1x MEM non-essential amino acid solution (Gibco, USA), 0.06% sodium bicarbonate (Gibco, USA), 20 μM HEPES (Gibco, USA) and 100 U/mL penicillin-streptomycin solution (Gibco, USA). Influenza A/Perth/265/2009 (H1N1)pdm09 virus was isolated from a clinical specimen at the World Health Organization Collaborating Centre for Reference and Research on Influenza (WHOCCRRI, Melbourne, Australia). Virus was plaque purified and propagated in MDCK cells.

## Ferrets and experimental groups

**London experiments.** Experiments were performed in a containment level 2 laboratory. Outbred female ferrets (20–24 weeks old) weighing 750–1000 g were used. Prior to the transmission study, ferrets were confirmed to be seronegative against the challenge virus strain, A/England/195/2009 (H1N1)pdm09, by haemagglutination-inhibition (HI) assay using turkey erythrocytes [16]. For inoculation, ferrets were lightly anesthetized with ketamine (22 mg/kg) and xylazine (0.9 mg/kg). All animals were nasal-washed daily, while conscious, by instilling 2 mL PBS into the nostrils, and the expectorate was collected in modified 250 mL centrifuge tubes. Ferrets were weighed daily post-infection, and body temperature was measured daily via subcutaneous IPTT-300 transponder (Plexx B.V, Netherlands). Bedding litter in the cage was changed daily. Ferret activity was scored manually at the same time every day (scoring was based on a previous publication [17]). Ferrets were observed whilst in their cages at least 4 hours post nasal washing in order to minimize disruption to their activity. For pharmacokinetic analysis, blood samples from baloxavir-treated ferrets were taken 48 and 144 hours after treatment and analyzed as described above.

The effect of BXA treatment on indirect contact (IC) transmission was assessed by employing an experimental setup that separated donor and IC sentinel ferrets by a minimum distance of 5 cm using a partition designed to eliminate direct contact (DC) but permit airflow exchange [18]. The experimental setup also included a DC component, where donor ferrets were concurrently co-housed with a DC sentinel ferret. Donor ferrets were inoculated intranasally with $10^4$ plaque forming units (pfu) of A/England/195/2009 (H1N1)pdm09 virus diluted in PBS (100 μL per nostril). Antiviral treatment of donors commenced 24 hours following

inoculation. BXA treatment of donors was compared with OST or an untreated control (n = 4 donors per group). Immediately after initiation of treatment, IC and DC sentinels were exposed to the same donor ferret (1:1:1 ratio) for 48 hours. At 14 days post infection (DPI), ferrets were injected with a non-reversible anesthetic consisting of ketamine ($\geq$25 mg/kg) and xylazine ($\geq$5 mg/kg), before sacrifice by intracardiac injection of sodium pentobarbitone ($\geq$1,000 mg/kg), at which time blood was collected by cardiac puncture. Endpoint sera were analyzed by HI assay.

**Melbourne experiments.** Experiments were performed in the Bio-Resources Facility (Physical Containment Level 2) at the Peter Doherty Institute (Melbourne, Australia). Outbred ferrets >12 weeks old and weighing 600–1800 g were used. Prior to transmission studies, ferrets were confirmed to be seronegative against recently circulating influenza A(H1N1)pdm09, A(H3N2) and B viruses by HI assay using turkey erythrocytes. Virus inoculation and subcutaneous drug delivery was performed under reversible anesthesia by intramuscular injection of ketamine (10 mg/kg, Troy Laboratories), midazolam (0.5 mg/kg, Troy Laboratories) and medetomidine (0.02 mg/kg, Troy Laboratories) mixture, which was then antagonized by atipamezole (0.01 mg/kg, Troy Laboratories). All animals were nasal washed daily by instilling 1 mL PBS into the nostrils to collect the expectorate. Nasal washes were performed under light sedation by intramuscular injection of xylazine (5 mg/kg, Troy Laboratories). Ferrets were weighed daily following infection/exposure, and body temperature was measured daily via subcutaneous microchip (LifeChip, Bio-Thermo). Bedding litter in the cage was changed daily.

The Melbourne DC transmission study consisted of three separate experiments using different schedules of donor antiviral treatment and timings of sentinel co-housing. Donor ferrets (n = 4 per treatment group) were inoculated intranasally with influenza A/Perth/265/2009 (H1N1)pdm09 virus at $10^3$ 50% tissue culture infectious dose (TCID$_{50}$) diluted in PBS (20 μL per nostril), following which antiviral treatment with either BXA, OST or placebo commenced at a designated time point. At a designated time following treatment, naïve sentinel ferrets were co-housed with treated donor ferrets (1:1) for a DC exposure period of 48 hours. Daily nasal wash samples were collected from 1 DPI until sacrifice for donor animals, and from 1 day post exposure (DPE) to 10 DPE for sentinel animals. At endpoint, ferrets were injected with a non-reversible anesthetic consisting of ketamine ($\geq$25 mg/kg) and xylazine ($\geq$5 mg/kg), before sacrifice by intracardiac injection of sodium pentobarbitone ($\geq$1,000 mg/kg, Troy Laboratories). Donors were sacrificed at the end of the 48 hour exposure period (2 DPE), and blood was collected from BXA- and placebo-treated ferrets by cardiac puncture for pharmacokinetic analysis (as described above). Sentinel ferrets were sacrificed at either 10 DPE (in the 24 hour treatment/co-housing experiment) or 16 DPE (in the 24 hour treatment/48 hour co-housing and 48 hour treatment/co-housing experiments). Blood was collected by cardiac puncture from sentinel ferrets which were sacrificed at 16 DPE, and the sera were analyzed by HI assay as previously described [17].

## Antiviral treatment of ferrets

BXA was delivered as a single dose subcutaneously at four locations on the dorsal region (4 mg/kg per animal) as described above. In the Melbourne experiments, control animals received a single placebo dose of methyl cellulose solution (1 mL/kg) alone delivered subcutaneously in the same way as BXA, whereas in the London experiments control ferrets were untreated. OST doses of 5 mg/kg were delivered twice daily (b.i.d.) (8h interval) for up to five days to non-sedated ferrets by providing it orally dropwise into the mouth of the ferret using a pipette (total dose 10 mg/kg/day). The OST dose was based on a previous pharmacokinetic analysis that showed a 5 mg/kg dose of oseltamivir every 12 hours for 5 days results in the

same median area under the plasma concentration-time curve drug concentration as observed in humans following the approved dose of 75 mg twice daily for 5 days [14].

## Virological analysis

**London experiments.** Immediately following collection each day, the nasal wash of ferrets was used for virus titration by plaque assay. The limit of virus detection in the plaque assays was 10 pfu/mL. Ferret nasal wash (140 μl) was also used to extract RNA using the Qiagen Viral RNA mini kit, according to manufacturer's instructions.

Quantitative real-time RT-PCR (qRT-PCR) was performed using 7500 Real Time PCR system (ABI) in 20 μl reactions using AgPath-ID One-Step RT-PCR Reagents 10 μl RT-PCR buffer (2X) (Thermo Fisher), 4μl of RNA, 0.8 μl forward (5'GACCRATCCTGTCACCTCTGA 3') and reverse primers (5' AGGGCATTYTGGACAAAKCGTCTA3') and 0.4 μl probe (5' FAM-TCGAGTCCTCGCTCACTGGGCACG-BHQ1 3'). The following conditions were used: 45˚C for 10 min, 1 cycle; 95˚C for 10 min, 1 cycle; 95˚C for 15 sec then 60˚C for 1 min, 40 cycles. For each sample, the $C_t$ value for the target M gene was determined. Based on the standard curves, absolute M gene copy numbers were calculated. Lower limit of quantification (LLOQ) was 153 M gene copies per μL RNA, based on the results from the samples of uninfected ferrets (mean + 2*standard deviation).

**Melbourne experiments.** For determination of infectious virus, nasal wash samples were stored with 1% w/v bovine serum albumin (BSA) at -80˚C before determination of viral titre by $TCID_{50}$ (LLOQ at 2.0 $\log_{10}$ $TCID_{50}$/mL) as previously described [17]. Viral RNA was extracted from 200 μL nasal wash samples using the NucleoMag VET isolation kit (Macherey Nagel) on the KingFisher Flex (ThermoFisher Scientific) platform according to manufacturer's instructions.

qRT-PCR was performed on the ABI 7500 Real Time PCR System (Applied Biosystems) using the SensiFAST Probe Lo-ROX One-Step qRT-PCR System Kit (Bioline) under the following conditions: 45˚C for 10 min, 1 cycle; 95˚C for 2 min, 1 cycle; 95˚C for 5 sec then 60˚C for 30 sec, 40 cycles. M gene-targeted universal influenza A real-time primer/probe sets (forward: 5'-GACCRATCCTGTCACCTCTGAC-3'; reverse: 5'-GGGCATTYTGGACAAAKCG TCTACG-3'; probe: 6FAM- TGCAGTCCTCGCTCACTGGGCACG -BHQ1) were kindly provided by CDC Influenza Branch (Atlanta, USA). Results were analyzed by 7500 Fast System SDS software v1.5.1. Influenza virus M gene copy number per μL RNA was determined by the standard curve method using influenza A RNA standards of known copy number, kindly provided by Seqirus (Melbourne, Australia). LLOQ was at 100 copies per μL RNA, based on first RNA standard to yield a $C_t < 35$.

## Sequencing

The emergence of viral escape mutants containing amino acid substitutions (such as PA/I38X) known to confer reduced susceptibility, was investigated by sequencing viruses from nasal washes of donor ferrets treated with BXA and sentinel ferrets by next generation sequencing (NGS) or pyrosequencing.

For NGS (London), viral RNA was extracted using the QIAamp Viral RNA mini kit (Qiagen). RNA was amplified using SuperScript III One-Step RT-PCR System with Platinum Taq DNA Polymerase and barcoded primers specific to PA (Forward primer ATGGAAGGCT TTGTGCGACA; barcoded reverse primers CATGTAACGCTCAATTATCTCAAATCGGTG, CATGGCTCTATCAATTATCTCAAATCGGTG, CATGATGGCTCAATTATCTCAAATC GGTG, and CATGCGATTGTCAATTATCTCAAATCGGTG). Library preparation was performed using NEBNext Ultra kit (NEB). Samples were sequenced giving 150 bp paired-end

reads on an Illumina MiSeq. Sequencing data for the samples were processed and analyzed using custom scripts in R. The sequence data can be found at https://www.ebi.ac.uk/ena under project number PRJEB33516. The code that was used to run these sequence analyses can be found at https://github.com/flu1. Plasmid spike-ins were used to validate the NGS.

For pyrosequencing (Melbourne), the MyTaq One-Step RT-PCR Kit was used to generate and amplify cDNA from viral RNA isolated from donor nasal wash samples. Samples were processed in the PyroMark Vacuum Workstation according to manufacturer's protocol, and pyrosequencing was performed using the PyroMark ID System (Biotage). Forward (Biotin-5'-CAATCCAATGATCGTCGAGC-3'), reverse (5'-GGTGCTTCAATAGTGCATTTGG-3') and pyrosequencing (5'-CAAACTTCCAAATGTGTGCA-3', reverse orientation) primers were designed to investigate SNPs at position 38 of the PA gene of influenza A(H1N1)pdm09 virus. Analysis was performed using PyroMarkQ96 ID Software to quantify the percentage of wild type and mutant in a mixed virus population.

## Statistical analyses

Data analysis was performed using GraphPad Prism (GraphPad Software, v5.01). Nasal wash viral titres of donor ferrets were log-transformed and then compared by two-way ANOVA followed by Bonferroni's post-tests. The area under the curve (AUC) for $TCID_{50}$/plaque assay, which estimated the total quantity of infectious virus shed for the duration of donor sampling, was compared by one-way ANOVA followed by Tukey's multiple comparison test. The mean time to first positive nasal wash was compared by Kruskal-Wallis test followed by Dunn's multiple comparison test. Samples below the LLOQ were assigned zero values for graphing and statistical analyses. qRT-PCR was performed on influenza control RNA of known copy number to generate a standard curve (linear regression of influenza cycle threshold [$C_t$] values vs. copy number). Viral copy numbers present in nasal wash RNA samples on the same assay were calculated using the formula of the standard curve. Influenza A copy numbers for each group of ferrets were log-transformed and then compared by two-way analysis of variance followed by Tukey's multiple comparisons test. $p < 0.05$ was considered statistically significant.

## Ethics statement

For the London experiments, all work performed was approved by the local genetic manipulation (GM) safety committee of Imperial College London, St. Mary's Campus (centre number GM77), and the Health and Safety Executive of the United Kingdom. Animal research described in the London experiments was carried out under a United Kingdom Home Office License, P48DAD9B4.

The Melbourne experiments were conducted with approval from the University of Melbourne Biochemistry & Molecular Biology, Dental Science, Medicine, Microbiology & Immunology, and Surgery Animal Ethics Committee (AEC#1714278), in accordance with the NHMRC Australian code of practice for the care and use of animals for scientific purposes (8th edition). For the Japanese pharmacokinetic studies, the animal study protocol was approved by the Shionogi Animal Care and Use Committee (# S18046C for BXA delivery and # S14089D for BXM delivery) in terms of the 3R (Replacement/Reduction/Refinement) principle.

## Results

### BXA pharmacokinetics in ferrets

Pharmacokinetic data from clinical trials show that the standard single oral dose of BXM in humans (40 mg for individuals <80 kg) results in a long half-life of 49–91 hours and plasma

concentrations of BXA (active form) that exceed the estimated target of effectiveness (6.85 ng/mL) for more than 120 hours, although PK profiles differ between Asian and non-Asian patients (Fig 1) [19,20]. The target plasma BXA concentration of ≥6.85 ng/mL was set from non-clinical studies in mice as the concentration necessary to exert greater virus reduction than that seen with oseltamivir phosphate. However, oral administration of 10 mg/kg BXM in ferrets resulted in rapid drug clearance within the first 24 hours [21] (Fig 1), resulting in the requirement for an alternative administration route. It was found that BXA prepared as a suspension (1 mg/mL) delivered by subcutaneous injection to four locations on the dorsal region as a single treatment (4 mg/kg in total) allowed maintenance of plasma BXA concentrations above the estimated target of effectiveness [19] for >144 hours following treatment (dotted line, Fig 1). This dose and method of BXA administration was therefore used for all subsequent experiments. Plasma samples taken from treated ferrets at various time points during the infection/treatment experiments demonstrated that the dosing achieved a level of drug exposure greater than 6.85 ng/mL (Fig 1).

## Baloxavir treatment of donor animals decreased virus shedding and alleviated clinical signs (London)

Ferrets infected with an A(H1N1)pdm09 virus were treated 24 hours post-infection with either BXA, OST or remained untreated (n = 4 per group) (Fig 2A). Analyses to evaluate the effect of antiviral treatment of donor animals on viral shedding involved 1) area under the curve

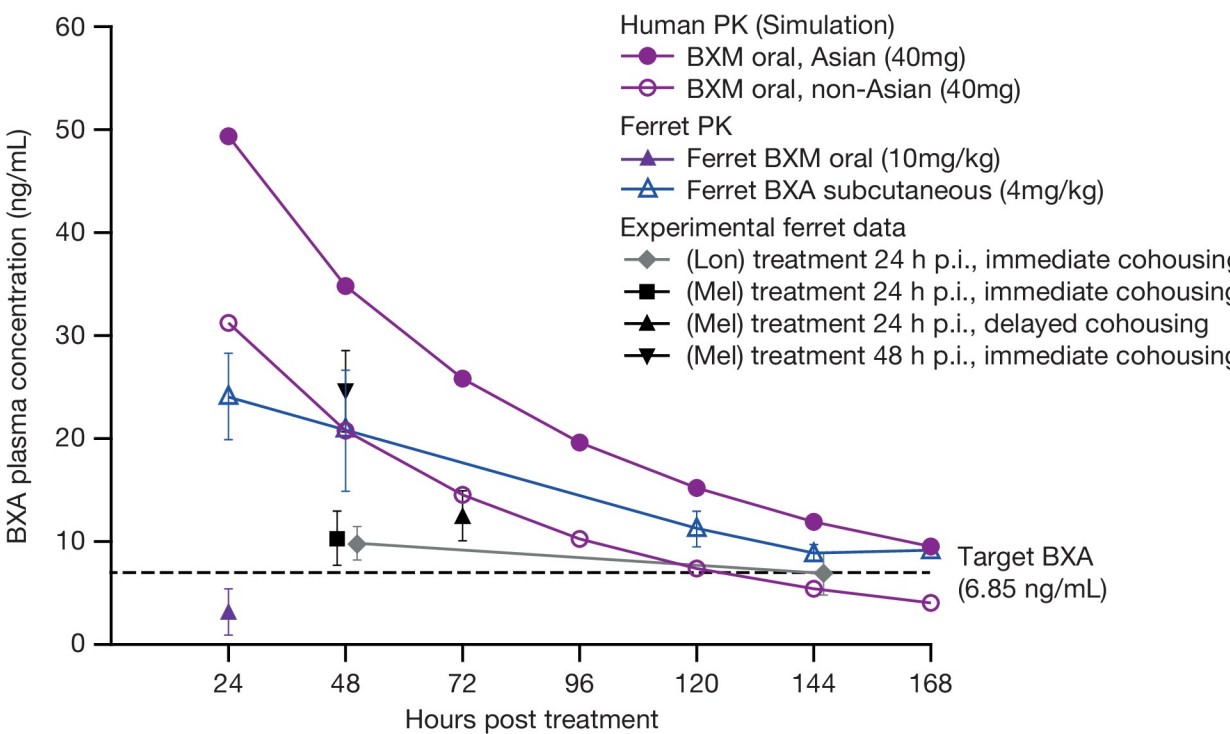

**Fig 1. Plasma concentrations (ng/mL + SD) of BXM and BXA in ferrets compared to humans.** Administration to ferrets was either via an oral (BXM) or subcutaneous (BXA) route. BXM was prepared at a concentration of 2 mg/ml and delivered to ferrets via an intragastric tube at a concentration of 10 mg/5 mL/kg. Subcutaneous injections were made using BXA in suspension (1 mg/mL) and administered to four sites on the dorsal region of the ferret (1 mL at each site). Human PK data was derived from population pharmacokinetics and exposure-response analyses in adults and adolescents [20]. Blood samples were collected at various time points and the plasma concentration of BXA was determined by LC-MS/MS. BXA-treated ferrets were sampled at 48 and 144 hours post-treatment in London (Lon) IC model experiments, or at 48 or 72 hours post-treatment in Melbourne (Mel) DC model experiments.

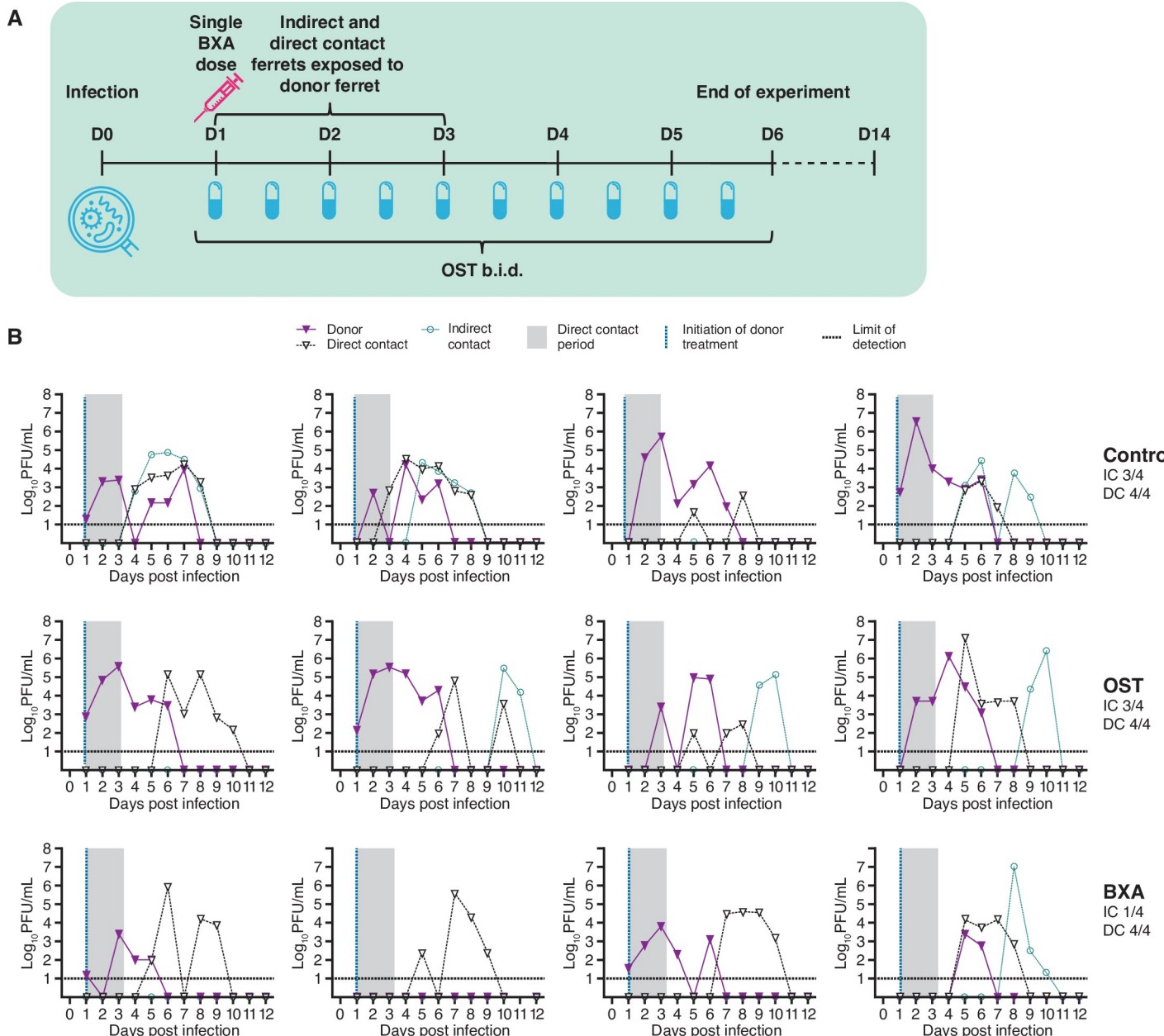

**Fig 2. Effect of BXA treatment on indirect transmission (London). (A) Experimental setup.** Donor ferrets were intranasally inoculated with $10^4$ PFU of A/England/ 195/2009. Antiviral treatment of infected donor ferrets commenced 24 hours post-infection. OST was administered a total of ten times across a five-day period; BXA was delivered as a single dose. Influenza-naïve sentinel DC ferrets were co-housed immediately following treatment. In addition, naïve sentinel IC ferrets were housed immediately after treatment in separate cages from those of the donor and DC sentinel ferrets. Nasal washes were collected from all donor and sentinel ferrets to assess shedding of infectious virus from 1 DPE to 11 DPE. **(B) Nasal wash infectious viral titres in donor and sentinel ferrets.** Donor ferrets were either untreated (upper panel), treated with oseltamivir (OST, middle) or treated with baloxavir (BXA, lower). Virus replication curves (plaque assay) for each donor and their corresponding DC and IC sentinels are graphed.

(AUC) analysis of plaque assay/TCID$_{50}$ data and 2) comparisons of mean viral titre at key time points. Infectious viral titres in nasal wash were significantly lower in BXA-treated ferrets at day 2 post-infection compared with untreated donor ferrets ($p = 0.043$), and at day 5 post-infection compared to both untreated and OST-treated donors ($p = 0.030$ and $p = 0.003$,

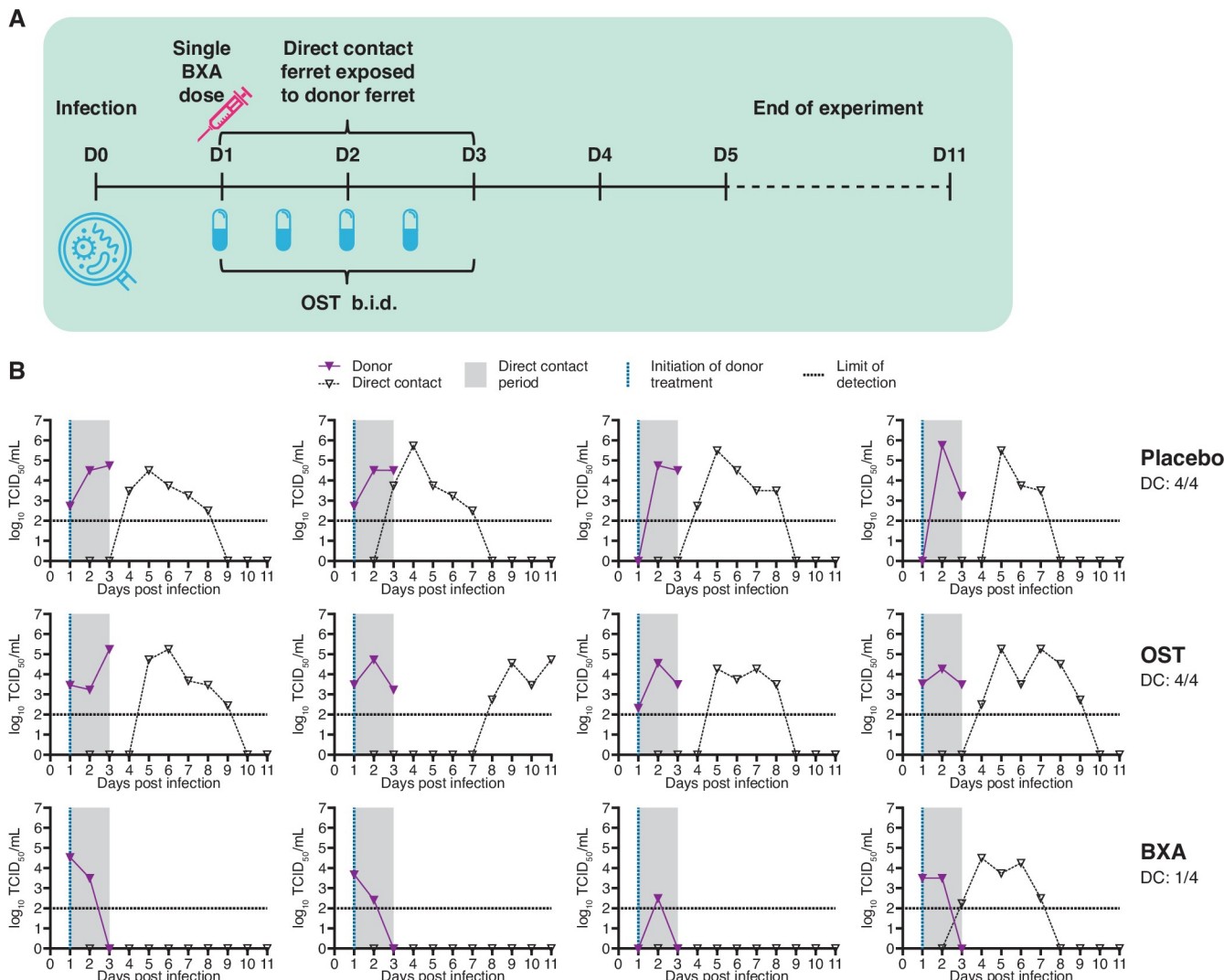

**Fig 3. Effect of BXA treatment on direct contact transmission after immediate co-housing of sentinel ferrets (Melbourne study). (A) Experimental setup.** Donor ferrets were intranasally inoculated with $10^3$ TCID$_{50}$ of influenza A/Perth/265/2009. Antiviral treatment of infected donor ferrets commenced 24 hours post-infection, and influenza-naïve sentinel ferrets were co-housed with donor ferrets immediately following treatment. OST was administered to donors a total of four times across a two-day period until sacrifice. BXA and placebo (methyl cellulose solution) were each delivered subcutaneously as a single dose. Nasal washes were collected from sentinel ferrets to assess shedding of infectious virus from 1 DPE to 10 DPE. A terminal bleed was not collected for this experiment. **(B) Nasal wash infectious viral titres in donor and sentinel ferrets.** Donor ferrets were either treated with placebo (upper panel), treated with oseltamivir (OST, middle) or treated with baloxavir (BXA, lower). Virus replication curves (TCID$_{50}$) for each donor/DC sentinel pair are graphed.

respectively) (Fig 2B). The AUC of infectious viral load over the entire course of infection was significantly less for the BXA treatment group (mean ± standard deviation, 7.07 ± 5.62) compared with the untreated (21.10 ± 5.60, $p = 0.014$) and oseltamivir groups (18.87 ± 5.17, $p = 0.033$). A similar effect was observed when viral titres were measured by qRT-PCR (S1A Fig) and corresponding HI antibody response data are presented in S1A Table. All donor ferrets were observed to be similarly active and displayed a fever at day 2 or 3 post-infection. No obvious weight loss was observed. Taken together, these data indicate that BXA dosing was effective in reducing viral load in treated animals.

## BXA treatment reduced IC transmission to ferrets exposed in an adjacent cage (London)

Analyses to evaluate the transmission of influenza virus from donor animals to sentinel animals involved 1) frequency of ferrets that became virus positive (by plaque assay/$TCID_{50}$ or RT-PCR on any day) or HI serology positive, and 2) time (days) to first virus positivity. Immediately following treatment at 24 hours post-infection, donor ferrets were exposed to naïve sentinel ferrets either in the same cage (to assess DC transmission) or in an adjacent separate cage (to assess IC transmission) for 48 hours. In the untreated control group, transmission occurred to all four DC sentinels (4/4) and to three of four (3/4) IC sentinels (based on infectious virus positivity) (Fig 2B, S1B Table). OST treatment of the donors had no effect on reducing the number of sentinel ferrets that became infected compared to placebo (4/4 DC sentinels and 3/4 IC sentinels). Although the mean time to first positive nasal wash in the DC sentinel animals was delayed in the OST treated and BXA treated ferrets (median 5.5 and 5 days, respectively) compared with the untreated ferrets (median 4.5 days), this difference was not statistically significant ($p = 0.15$). However, BXA did reduce frequency of transmission of virus to IC sentinels, where only 1/4 sentinels became infected, based on infectious virus, qRT-PCR or HI serology (Fig 2B, S1 Table).

In this experimental setup, BXA treatment of infected ferrets was able to reduce IC transmission but not DC transmission, suggesting that blocking DC transmission presents a more stringent challenge than blocking transmission by the IC route. Using different experimental conditions in the Melbourne laboratory, we further explored the potential for BXA treatment to reduce DC transmission.

## BXA treatment 24 hours post-infection reduces DC transmission to ferrets exposed from the start of treatment (Melbourne)

The effect of antiviral treatment on transmission was assessed in a DC model where BXA, OST or placebo were administered to donors 24 hours after infection with an A(H1N1)pdm09 virus, followed by immediate exposure (co-housing) of naïve sentinels (Fig 3A). Similar to the London experiment, BXA treatment at this time point resulted in a significant reduction of infectious viral shedding compared to both OST and negative control (placebo) treatment. Nasal wash infectious virus titres in BXA-treated donors were lower on day 2 ($p{\leq}0.05$) compared with placebo, and on day 3 ($p{\leq}0.001$) post-infection compared with both placebo and OST (Fig 3B). All placebo- and OST-treated ferrets were shedding infectious virus in nasal washes at the end of the 48 hour exposure period, compared with none of the BXA-treated ferrets (Fig 3B). For the duration of donor sampling, the infectious viral load (group mean AUC ± SD) showed that BXA-treated ferrets (4.47 ± 1.43) shed significantly less virus in nasal washes than either OST- (7.73 ± 0.32, $p{\leq}0.01$) or placebo-treated donors (7.69 ± 0.60, $p{\leq}0.01$). Infected ferrets exhibited fever from 1 day post-infection. No substantial weight loss was observed in OST- or BXA-treated ferrets, but the body weight of placebo-treated ferrets was significantly lower than BXA-treated ferrets at 3 days post-infection ($p{\leq}0.05$).

Nasal washes of co-housed DC sentinels showed that OST treatment of donors did not reduce transmission compared to placebo, as infectious virus shedding was detected in 4/4 sentinels for both groups. However, in contrast to the DC arm of the London experiment, transmission to ferrets co-housed with BXA-treated donors was reduced, as infectious virus was detected in only 1/4 sentinels following 48 hours of exposure (Fig 3B, S1 Table). We did not observe significant differences between groups in the time to detection of virus positivity based on $TCID_{50}$. Based on qRT-PCR analysis, transient shedding of viral RNA was detected near the LLOQ in an additional DC sentinel in the BXA group for one day, though this sample

did not cause detectable infection in cell culture (S1B Fig). HI antibody titres were not available from this experiment.

## BXA treatment 24 hours post-infection reduces DC transmission to ferrets after delayed exposure (Melbourne)

In the subsequent experiment, donors were treated 24 hours after infection with an A(H1N1) pdm09 virus, but the co-housing of naïve DC sentinels was delayed until 24 hours following treatment (Fig 4A). Consistent with previous observations, BXA treatment at this time point resulted in a significant reduction in infectious viral titres on days 2 ($p \leq 0.05$), 3 ($p \leq 0.01$) and 4 ($p \leq 0.001$) post-infection compared with placebo, and on days 3 ($p \leq 0.01$) and 4 ($p \leq 0.001$) post-infection compared to OST (Fig 4B). Placebo-treated and OST-treated donor ferrets displayed similar viral replication kinetics, and 4/4 ferrets in each of these treatment groups were still shedding infectious virus at the end of the 48 hour contact exposure period (Fig 4B). In contrast, no BXA-treated donors shed detectable infectious virus at this time point (Fig 4B), and BXA-treatment significantly reduced infectious viral load during the duration of sampling (AUC 6.34 ± 1.44) compared to both OST (10.22 ± 2.38, $p \leq 0.05$) and placebo (12.35 ± 0.53, $p \leq 0.01$). Infected ferrets exhibited fever from 2 days post-infection. No substantial weight loss was observed in OST- or BXA-treated animals, but body weight for placebo-treated ferrets was significantly lower than OST-treated animals at 3 days post-infection ($p \leq 0.05$), and BXA-treated ferrets at 4 days post-infection ($p \leq 0.01$).

Despite a 24 hour delay in exposure, the incidence of DC transmission from OST-treated donors remained identical to the placebo group by infectious virus shedding, with 4/4 co-housed sentinels shedding detectable infectious virus after 48 hours of co-housing (Fig 4B). Once again, BXA-treatment resulted in a reduction in transmission compared to OST and placebo, as infectious virus was detected in only 1/4 DC ferrets co-housed with BXA-treated donors. We did not observe significant differences between groups in the time to detection of virus positivity based on $TCID_{50}$. qRT-PCR analysis indicated the same reduction in transmission frequency in the BXA group (S1 Fig); however, a low HI antibody titre was detected in one additional BXA DC ferret (2/4 overall) which had tested virus negative by $TCID_{50}$ and qRT-PCR (S1 Table).

## Delayed treatment with BXA at 48 hours post-infection reduces DC transmission to sentinel ferrets (Melbourne)

In the final DC experimental setup, BXA treatment of donor ferrets was delayed until 48 hours post-infection and DC ferrets were co-housed with donors immediately after treatment (Fig 5A). In BXA-treated donor ferrets, a significantly lower infectious virus titre was observed on 3 DPI (1 day after treatment) compared to placebo ($p \leq 0.001$) and on 4 DPI compared to both placebo and OST treatment ($p \leq 0.001$) (Fig 5B). Compared to placebo, OST treatment significantly reduced infectious virus on 3 DPI only ($p \leq 0.05$). Despite the delay in drug administration, as in previous experiments there was no detectable infectious virus in any of the BXA-treated donor ferrets at the end of the exposure period (2 DPE), whereas at the same time point, all OST- and placebo-group donors were still shedding infectious virus. For the duration of sampling, the infectious viral load shed by BXA-treated ferrets (AUC 8.65 ± 1.70) was significantly less than placebo (13.06 ± 0.94, $p \leq 0.01$). Infected ferrets exhibited fever from 1 or 2 days post-infection. All groups exhibited mild weight loss, but no significant differences in body weight were observed between groups.

Treatment of donor ferrets with BXA 48 hours post-infection prevented onward transmission of virus to 2/4 co-housed sentinel ferrets based on virus titration, compared to OST- or placebo-treated donors where DC transmission occurred to all exposed sentinel ferrets (Fig

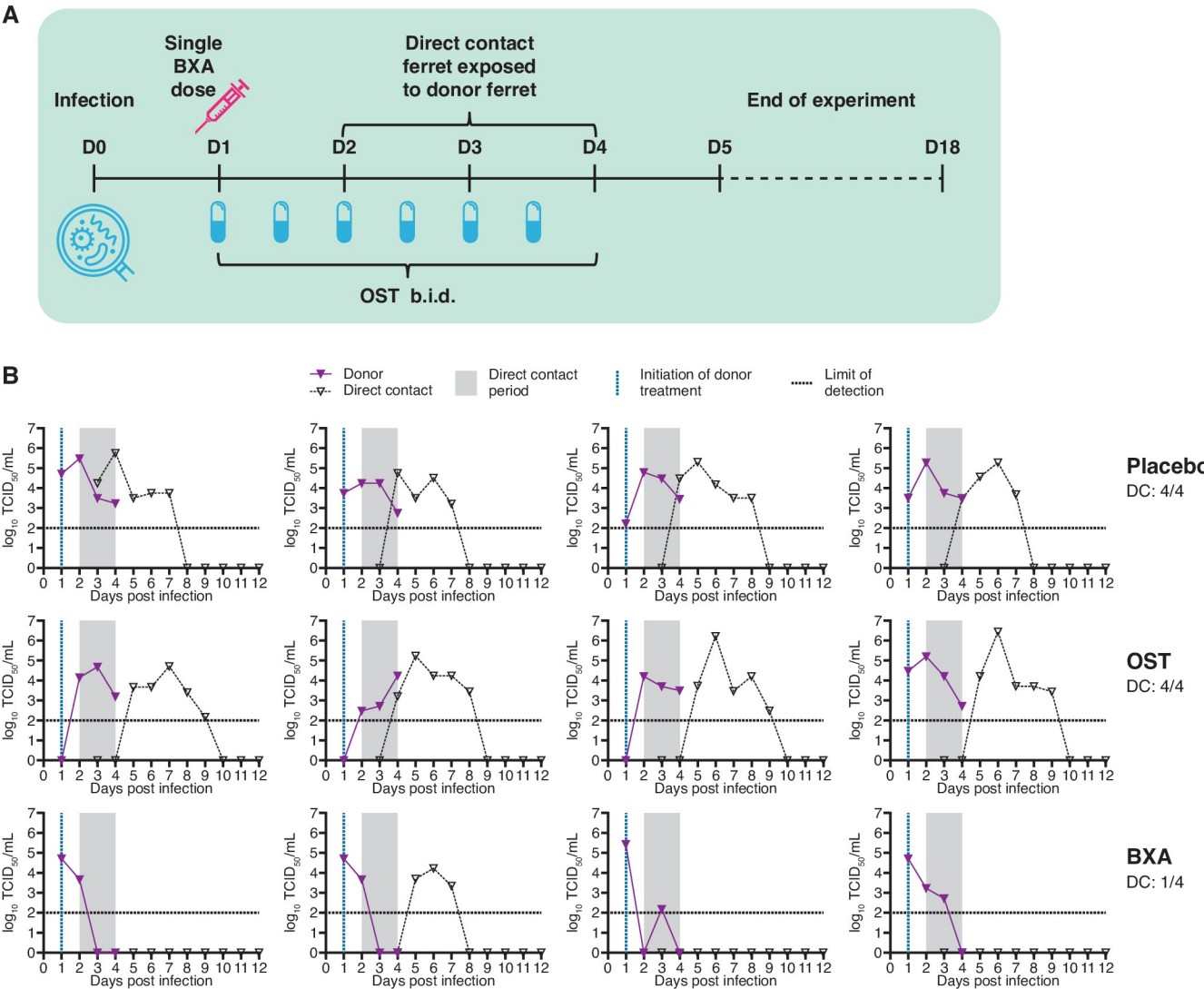

**Fig 4. Effect of BXA treatment on direct contact transmission after delayed co-housing of sentinel ferrets (Melbourne study). (A) Experimental setup.** Donor ferrets were intranasally inoculated with $10^3$ $TCID_{50}$ of influenza A/Perth/265/2009. Antiviral treatment of infected donor ferrets commenced 24 hours post-infection, and influenza-naïve sentinel ferrets were co-housed with donor ferrets 24 hours post-treatment. OST was administered to donors a total of six times across a three-day period until sacrifice. BXA and placebo (methyl cellulose solution) were each delivered subcutaneously as a single dose. Nasal washes were collected from sentinel ferrets to assess shedding of infectious virus from 1 DPE to 10 DPE, and a terminal bleed was collected for serology at 16 DPE. **(B) Nasal wash infectious viral titres in donor and sentinel ferrets.** Donor ferrets were either treated with placebo (upper panel), treated with oseltamivir (OST, middle) or treated with baloxavir (BXA, lower). Virus replication curves ($TCID_{50}$) for each donor/DC sentinel pair are graphed.

5B). We did not observe significant differences between groups in the time to detection of virus positivity based on $TCID_{50}$. Using qRT-PCR, it was observed that 3/4 sentinel ferrets in the BXA treatment group were virus positive (S1D Fig) and 4/4 were positive based on serological analysis, although the two ferrets that were infectious virus negative had only low HI antibody titres (S1 Table).

## No viruses with PA/I38X amino acid substitutions were detected

An isoleucine-to-threonine substitution at amino acid position 38 of the PA gene (PA/I38T) has been shown to confer reduced susceptibility to BXA [22,23]. Based on pyrosequencing or

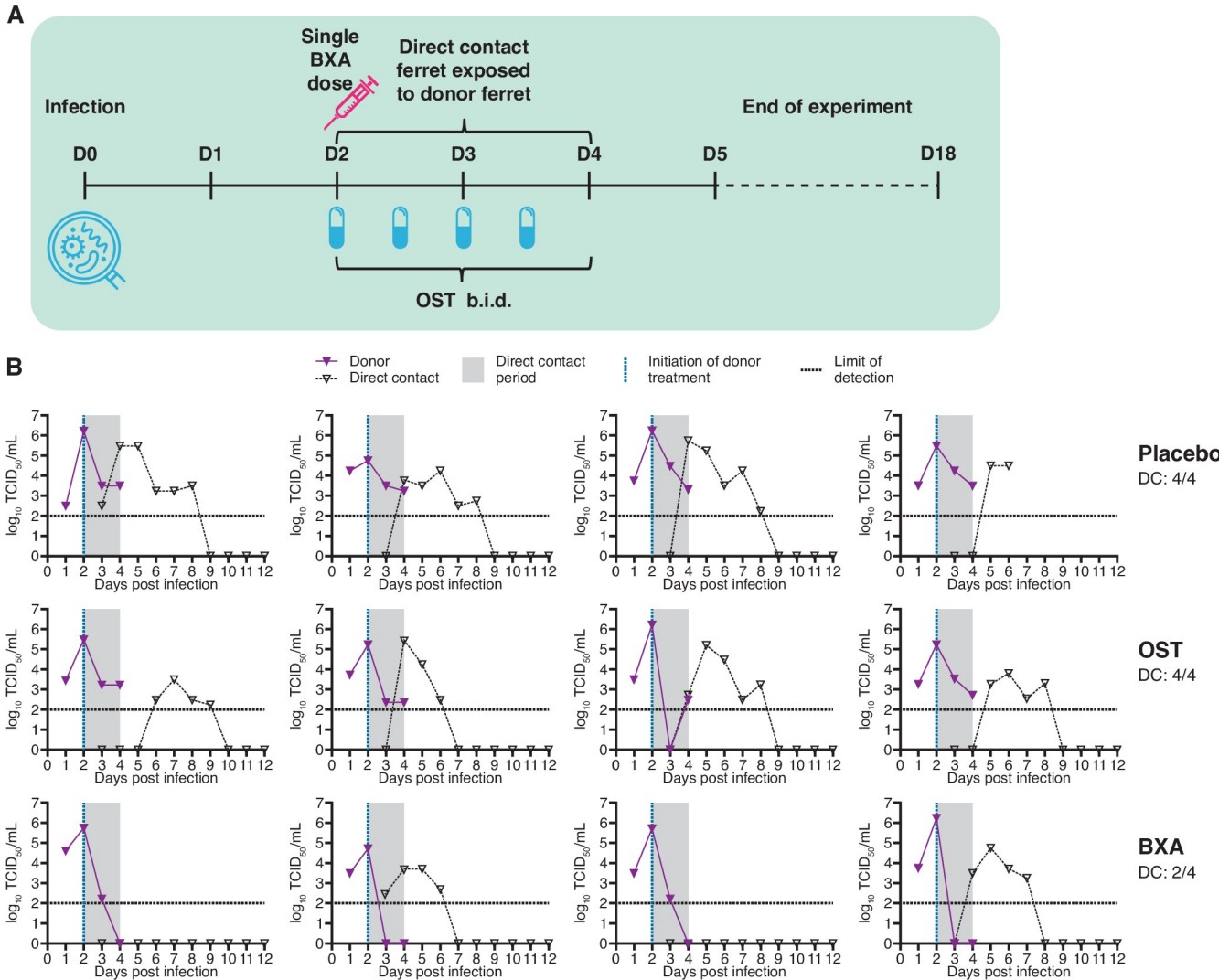

**Fig 5. Effect of delayed BXA treatment on direct contact transmission (Melbourne study). (A) Experimental setup.** Donor ferrets were intranasally inoculated with $10^3$ $TCID_{50}$ of influenza A/Perth/265/2009. Antiviral treatment of infected donor ferrets commenced 48 h post-infection, and influenza-naïve sentinel ferrets were co-housed with donor ferrets immediately following treatment. OST was administered to donors a total of four times across a two-day period until sacrifice. BXA and placebo (methyl cellulose solution) were each delivered subcutaneously as a single dose. Nasal washes were collected from sentinel ferrets to assess shedding of infectious virus from 1 DPE to 10 DPE, and a terminal bleed was collected for serology at 16 DPE. **(B) Nasal wash infectious viral titres in donor and sentinel ferrets.** Donor ferrets were either treated with placebo (upper panel), treated with oseltamivir (OST, middle) or treated with baloxavir (BXA, lower). Virus replication curves ($TCID_{50}$) for each donor/DC sentinel pair are graphed.

NGS analysis of all ferret nasal wash samples obtained in both London and Melbourne, no A (H1N1)pdm09 virus variants with PA/I38X substitution were detected in BXA-treated donor ferrets or in sentinel ferrets exposed to the donors.

## Discussion

The approval of baloxavir for the treatment of influenza in otherwise-healthy patients in Japan, USA and other countries in 2018/19, heralded the first influenza antiviral with a new mode of action to be licensed in nearly 20 years. In clinical trials, a single dose of BXM was shown to have a rapid and potent virological effect, reducing the median time of virus shedding by 3 days compared to placebo-treated patients and by 2 days compared to OST-treated

patients. Here we describe successful recapitulation of the robust virological effect of baloxavir that was seen in humans using the ferret model, and we demonstrate how the reduction in viral shedding leads to reduced transmission to contacts.

In our study, we were able to significantly reduce infectious viral titres in ferrets using subcutaneous administration of BXA. Although it would have been ideal to replicate human oral dosing using the prodrug BXM, the rapid clearance of orally delivered drug in ferrets did not match the human pharmacokinetic (PK) profile (Fig 1). However, subcutaneous delivery of 4 mg/kg BXA to ferrets resulted in a PK profile that was similar to that seen in humans who had received a 40 mg oral dose (Fig 1). Under this BXA regimen, we observed a significant reduction in viral titres compared to placebo at 24 hours following treatment, whereas oral OST treatment had little or no effect on infectious viral shedding. Despite using an unconventional drug delivery method, our observations in the ferret model were similar to human studies utilising the licensed prodrug. In a recent clinical trial, oral BXM treatment resulted in a 3.5 $\log_{10}$ $TCID_{50}$/ml reduction in viral shedding at 24 hours post-treatment, whereas OST achieved only a 1.5 $\log_{10}$ $TCID_{50}$/ml reduction [6]. Previous reports have similarly demonstrated that baloxavir has a greater antiviral effect than OST: a network meta-analysis of past randomized controlled trial data showed that baloxavir-treatment was significantly more effective in reducing infectious viral tires at 24 hours after administration than OST, and other NAIs including zanamivir and peramivir [24].

The superior antiviral effect of BXA treatment resulted in a downstream effect of limiting onward transmission of influenza viruses to naïve ferrets, both when they were housed in separate adjacent cages, and also when they were co-housed in a direct contact model of transmission. In contrast, OST treatment of donor ferrets had no effect on the likelihood of infecting sentinel animals compared to placebo in either IC or DC scenarios. As infectious viral loads in the nasal washes of donor animals were consistently lowest for BXA-treated animals in each experiment, the corresponding sentinels experienced lower exposure to virus compared to OST- or placebo-group sentinels during the 48 hour co-housing period. In the context of antiviral interventions, there is some evidence that more efficient reduction of viral shedding in infected index cases correlates with a lower likelihood of transmission to susceptible hosts. Large-scale human studies have reported modest [9] or non-significant [8,25] reductions in the secondary attack rates (SAR) of laboratory-confirmed influenza amongst household contacts of OST-treated index patients. However, zanamivir and peramivir, which can more effectively limit viral shedding compared to OST [11,26,27], have shown greater benefit in reducing household transmission than OST in observational studies [11,28,29]. There is limited experimental evidence to recapitulate this effect in a relevant animal model. A previous ferret study reported that oral OST (5 mg/kg b.i.d.) treatment of index animals initiated at 36 hours post-infection did not significantly reduce nasal wash shedding of influenza A(H1N1)pdm09 virus, and did not reduce the incidence of transmission to co-housed untreated contacts [30]. In a subsequent study, OST therapy was able to reduce infectious viral shedding in ferrets inoculated with pdm09 and avian H7N3 viruses by the ocular-only aerosol route, and consequently reduced direct contact transmission to co-housed contacts [31]. Our results in the ferret model support the hypothesis that the potent antiviral effect of BXA can efficiently interrupt the onward transmission of influenza viruses to untreated contacts.

In this study, two variables were tested: the time of antiviral treatment (24 to 48 hours post-infection) and the time between treatment and co-housing with naïve sentinel ferrets. With respect to the time of co-housing, we hypothesized that a delay in the exposure of sentinels until 24 hours after treatment would provide a greater chance for the antiviral drug to take effect compared with placebo. However, we found that the effect of BXA treatment was rapid and therefore a reduction in transmission frequency was observed regardless of whether ferrets

were co-housed immediately or 24 hours after treatment. Many studies have observed that early administration of influenza antivirals is associated with better clinical outcome and enhances the impact on viral shedding [32–35]. Our results demonstrate that delayed antiviral treatment (from 24 hours to 48 hours post-infection) was less effective in reducing viral transmission than early antiviral treatment. In this scenario, infectious viral shedding was detected in 2/4 BXA-group sentinels, but the other animals demonstrated evidence of viral exposure by qRT-PCR and serology (S1 Table). However, compared to culture-positive infections, HI titres (S1 Table) and viral RNA loads (S1 Fig) were very low. In a household setting, the clinical relevance of such mild infections is uncertain as such individuals are unlikely to develop acute symptoms requiring medical care, and further transmission is improbable in the absence of infectious viral shedding [36]. Taken together, our results show that BXA treatment remained effective in reducing transmission up to 48 hours after infection. This is important because in reality an influenza infected patient is more likely to seek treatment at 48 hours after infection when clinical symptoms have typically developed, rather than during the incubation period of the virus immediately following infection. The timeliness of antiviral treatment in index patients has been reported to modulate SAR in observational household studies. Primary case-patients receiving OST within 48 hours of symptom onset were less likely to transmit A (H1N1)pdm09 virus to household contacts compared to those treated beyond this time [37], and a similar effect was observed amongst zanamivir-treated index patients in a separate study [11].

The data generated from influenza virus transmissibility studies in ferrets can differ between laboratories due to heterogeneity among the outbred animals and in the experimental setups. This can create difficulties for direct comparisons or the combination of data from different studies [38]. The ability of BXA treatment to reduce onward transmission in the DC model was observed consistently in the three experiments from the Melbourne laboratory, but was not observed in the experiments from the London laboratory. This difference is likely to be due to greater DC transmission efficiency in the London laboratory setup, making it harder for an antiviral intervention to have an effect. Several potential factors may be responsible for this difference in DC transmission efficiency [18,38,39], but the most likely reasons for the enhanced antiviral effect seen in the Melbourne experiments include higher airflow in cages and lower volume and infectious doses used to infect ferrets (40 μl at $10^3$ TCID$_{50}$ vs 200 μl at $10^4$ PFU). Other possible reasons include differences in the viruses (a reverse genetics derived recombinant strain was used in the London experiments whilst a cell culture isolate of a naturally circulating virus was used in the Melbourne experiments), as well as differences in the ferrets and ambient conditions of the experiment (i.e. temperature and humidity), all of which may influence transmission dynamics [18,38]. Therefore, comparisons made within (rather than between) ferret studies remain the most reliable. In each study a significant impact of BXA treatment compared to OST or placebo/untreated on DC or IC transmission was demonstrated.

In clinical studies, the emergence of PA/I38X amino acid substituted viruses with reduced BXA susceptibility has been observed at frequencies ranging from 2.2% (4/182) in adults and adolescents to 23.4% (18/77) in children [5,6,40]. The lack of detection of virus variants with the PA/I38X mutation in the ferrets from the Melbourne experiments may have been due to the shorter sampling period (treated ferrets were culled at 2 to 3 days post-treatment). However, PA/I38X variants were not observed in the London experiment either, where there was an extended sampling period. In humans, a higher frequency of resistance has been reported in A(H3N2) viruses compared to A(H1N1)pdm09 viruses [41], and therefore it is possible that resistant variants may have emerged in treated ferrets if an A(H3N2) virus was used.

In conclusion, the data generated from the ferret models provides the first evidence that the rapid reduction in infectious viral titre associated with BXA treatment translates into a reduced risk of transmitting influenza to exposed contacts. Encouragingly, BXA remained effective at reducing transmission even when treatment was delayed until 2 DPI. In contrast, OST treatment had no effect on reducing secondary transmission. A first-of-its-kind phase III clinical trial (CENTERSTONE) is currently underway to investigate whether baloxavir treatment of influenza-infected index cases can reduce transmission of influenza to household contacts (NCT03969212). If efficacy is shown, baloxavir would be the first antiviral to demonstrate a dual effect of reducing symptoms in treated patients, and reducing transmission to contacts. Such an effect has the potential to dramatically change how we manage influenza outbreaks, including pandemic influenza.

## Supporting information

**S1 Fig. Nasal wash viral loads by quantitative real-time PCR in the absence or presence of antiviral treatment.** Copies of the influenza A M gene per μL RNA are displayed. London: Donor ferrets were intranasally inoculated with $10^4$ PFU of influenza A/England/195/2009. Melbourne: Donor ferrets were inoculated with $10^3$ $TCID_{50}$ of influenza A/Perth/265/2009 by the intranasal route. (A) (London) donor ferrets receiving no treatment (upper panel), OST (middle) or BXA (lower) at 24 hours p.i. were immediately exposed to naïve DC and IC sentinels. Log-transformed graphs are shown. (B) (Melbourne) donors treated at 24 hours p.i. were immediately co-housed with naïve DC sentinels. (C) (Melbourne) donors treated with placebo (upper), OST (middle) or BXA (lower) at 24 hours p.i. were co-housed with naïve DC sentinels 24 hours later (D) (Melbourne) donors treated with antivirals at 48 hours p.i. were immediately co-housed with naïve DC sentinels.
(TIF)

**S1 Table. Detection of A(H1N1)pdm09 virus infection in each individual sentinel animal by viral culture, qRT-PCR and serum antibody response.**
(DOCX)

## Acknowledgments

The authors thank staff at the Peter Doherty Institute Bioresources Facility for their expertise and assistance. Editorial support was provided by Gardiner-Caldwell Communications (GCC), Macclesfield, UK.

## Author Contributions

**Conceptualization:** Steffen Wildum, Neil Collinson, Klaus Kuhlbusch, Barry Clinch, Aeron C. Hurt, Wendy S. Barclay.

**Formal analysis:** Leo Yi Yang Lee, Jie Zhou, Daniel H. Goldhill.

**Funding acquisition:** Neil Collinson.

**Investigation:** Leo Yi Yang Lee, Jie Zhou, Rebecca Frise, Paulina Koszalka, Edin J. Mifsud, Kaoru Baba, Takahiro Noda, Yoshinori Ando, Kenji Sato, Aoe-Ishikawa Yuki, Elke Zwanziger.

**Methodology:** Leo Yi Yang Lee, Jie Zhou, Rebecca Frise, Daniel H. Goldhill, Paulina Koszalka, Edin J. Mifsud, Kaoru Baba, Takahiro Noda, Yoshinori Ando, Kenji Sato, Aoe-Ishikawa Yuki, Takao Shishido, Elke Zwanziger, Aeron C. Hurt.

**Writing – original draft:** Leo Yi Yang Lee, Jie Zhou, Steffen Wildum, Aeron C. Hurt, Wendy S. Barclay.

**Writing – review & editing:** Leo Yi Yang Lee, Jie Zhou, Rebecca Frise, Daniel H. Goldhill, Paulina Koszalka, Edin J. Mifsud, Kaoru Baba, Takahiro Noda, Yoshinori Ando, Kenji Sato, Aoe-Ishikawa Yuki, Takao Shishido, Takeki Uehara, Steffen Wildum, Elke Zwanziger, Neil Collinson, Klaus Kuhlbusch, Barry Clinch, Aeron C. Hurt, Wendy S. Barclay.

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
