## [Decision Letter · Decision Letter 0]

5 Jan 2020

Dear Dr. Barclay,

Thank you very much for submitting your manuscript "Baloxavir treatment of ferrets infected with influenza A virus reduces onward transmission" (PPATHOGENS-D-19-02155) for review by PLOS Pathogens. Your manuscript was fully evaluated at the editorial level and by independent peer reviewers. The reviewers appreciated the attention to an important problem and were supportive of publication in PLOS Pathogens. Nevertheless, each reviewer raised a number of important concerns about the manuscript as it currently stands. The reviewers viewed the inclusion of data from both London and Melbourne as a clear strength, but also noted inconsistencies in the reporting of data and methodologies from each site. Problems were also highlighted with the validity of comparison between baloxavir and oseltamivir. More generally, a number of valid concerns with clarity and presentation were also raised. These issues must be addressed before we would be willing to consider a revised version of your study. We cannot, of course, promise publication at that time.

We therefore ask you to modify the manuscript according to the review recommendations before we can consider your manuscript for acceptance. Your revisions should address the specific points made by each reviewer.

(1) A letter containing a detailed list of your responses to the review comments and a description of the changes you have made in the manuscript. Please note while forming your response, if your article is accepted, you may have the opportunity to make the peer review history publicly available. The record will include editor decision letters (with reviews) and your responses to reviewer comments. If eligible, we will contact you to opt in or out.

(2) Two versions of the manuscript: one with either highlights or tracked changes denoting where the text has been changed; the other a clean version (uploaded as the manuscript file).

Additionally, to enhance the reproducibility of your results, PLOS recommends that you deposit your laboratory protocols in protocols.io, where a protocol can be assigned its own identifier (DOI) such that it can be cited independently in the future. For instructions see http://journals.plos.org/plospathogens/s/submission-guidelines#loc-materials-and-methods

We hope to receive your revised manuscript within 60 days. If you anticipate any delay in its return, we ask that you let us know the expected resubmission date by replying to this email. Revised manuscripts received beyond 60 days may require evaluation and peer review similar to that applied to newly submitted manuscripts.

[LINK]

Sincerely,

Anice C. Lowen

Associate Editor

PLOS Pathogens

Ron Fouchier

Section Editor

PLOS Pathogens

Kasturi Haldar

Editor-in-Chief

PLOS Pathogens

orcid.org/0000-0001-5065-158X

Michael Malim

Editor-in-Chief

PLOS Pathogens

orcid.org/0000-0002-7699-2064

Reviewer's Responses to Questions

**Part I - Summary**

Reviewer #1: In this well-written article, authors present pharmacokinetic data from ferrets treated with baloxavir, and demonstrate that one treatment of BXA in influenza virus-inoculated ferrets reduces viral load in the upper respiratory tract and leads to a reduced frequency of virus transmission to contact animals compared with multi-day oseltamivir treatment, with no emergence of resistant viruses. The manuscript is notable in that these findings are demonstrated in two separate research groups, using two separate transmission models, challenge viruses, and experimental designs, supporting that these data are reproducible across different experimental conditions. Experiments are well-controlled, data is generally clearly presented, and results will be of interest to the field. However, there are areas of the manuscript (notably with regard to harmonization between presentation of two different methodologies employed, and clarity in some presentation of results and experimental conditions) that would benefit from additional attention.

Reviewer #2: In 2018, the arsenal of antiviral treatment options for influenza was extended with an approval in the United States and Japan of Baloxavir marboxil (BXM), the influenza cap-dependent endonuclease inhibitor. One of the characteristic features of BXM is rapid reduction of influenza virus shedding in the upper respiratory tract of treated patients. However, the implication of rapid reduction of virus load in the upper respiratory tract on transmissibility was unknown. In this study the authors addressed the question whether BXM treatment can affect transmissibility to contacts of influenza A(H1N1)pdm09 virus in ferrets, the gold standard animals for evaluating transmission. Ferret transmission experiments were conducted in two independent laboratories: Imperial College London, London, UK and The Peter Doherty Institute for Infection and Immunity, Melbourne, Australia.

Reviewer #3: The manuscript submission by Wendy S. Barclay and colleagues reports on the efficacy of Baloxavir therapy to prevent transmission of influenza viruses among ferrets. A strength of the study is the independent assessment of Baloxavir therapy at two differet reserach institutions. The authors report the pharmacokinetic (PK) profile of Baloxavir in ferrets. Treatment with Baloxavir reduced transmission among ferrets; however, Oseltamivir thereapy was less efficacious in reducing transmission of influenza virus among ferrets. There are two major weaknesses to the study. Ferrets were delivered the active form of Baloxavir, which is immediately active against influenza viruses; however, ferrets were given oral administration of Oseltamivir phosphate, which must be metabolized to the active form, oseltamivir carboxylate. The other weakness to the study is that the PK profile of Oseltamivir was not assessed following treatment of ferrets. Thus, there is not an equal comparison between the efficacy of Baloxavir versus Oseltamivir.

**Part II – Major Issues: Key Experiments Required for Acceptance**

Reviewer #1: (No Response)

Reviewer #2: 1. The experimental design and conditions of conducting experiments in a ferret model in two different laboratories were different. Importantly, the obtained results also differ in two different laboratories, as in London (where both DC and IC route of transmission were studied), the author did not see effect of BXA treatment on A(H1N1)pdm09 virus transmission to DC ferrets. In Melbourne (where only DC route of transmission was studied), the authors saw the effect of BXA treatment on A(H1N1)pdm09 virus transmission to DC ferrets. The authors discussed the factors affecting the obtained results. However, they must also include that in London recombinant influenza A(H1N1)pdm09 virus was used and in Melbourne – virus isolated from patient. At the beginning of Results section, the authors should indicate criteria used to conclude about the virus transmissibility to DC and IC ferrets. For example, the following criteria were used: 1) AUC of infectious virus load over entire course of infection, 2) mean time to first virus-positive nasal wash; 3) frequency of transmission of virus. The same criteria must be used for analysis of the data in both laboratories.

2. The data on clinical signs of infection was not presented and was not described. However, the authors were monitoring weight, temperature and activity of ferrets. Please provide a brief description of the obtained results.

3. The influenza virus titers in nasal washes of ferrets are presented in individual animals (n = 4/group, Figs. 2 - 5). It will be helpful for the readers if mean virus titers ± SD will be provided for each treatment group (control, OST, BXA) and statistical differences will be indicated.

4. Abstract must be modified. The current Abstract contains well-known information about influenza vaccines and antiviral drugs (half of the body of the Abstract) and very brief description of obtained results. PK data was not included. The Author summary must be modified – only two last sentences of summary are related to the current study.

Reviewer #3: The authors should address the PK profile of Oseltamivir in ferrets, especailly for the test regimen described in the current study, which is not the standard of patient care. The authors should also revise statements pertaining to the enhanced thereapeutic effect of Baloxavir over Oseltamivir since ferrets were treated with the active form of Baloxavir but the non-active form of Oseltamivir.

**Part III – Minor Issues: Editorial and Data Presentation Modifications**

Reviewer #1: 1. The methods section states that administration of both BXM and BXA were evaluated in the ferret model (line 110), but methodology is only provided for BXA evaluation. Considering BXM data following oral delivery is presented in Figure 1, the authors should include additional information in the methods regarding BXM evaluation (dose/volume/diluent for oral evaluation).

2. Interspersed throughout the methods and text, there are places where information is not uniformly described between London and Melbourne-based experiments, leading to potential confusion by readers regarding which properties are/are not similar or divergent. In example, line 154 (London) states that pre-inoculation sera was only tested against homologous virus whereas line 180 (Melbourne) states that pre-inoculation sera was screened against “recently circulating” influenza viruses; does this mean that London ferrets were not similarly screened? Similarly, Melbourne-based, but not London-based, experiments specify freezing ferret nasal samples prior to titration; does this imply that London samples were titered immediately post-collection without a freeze/thaw cycle?

3. The authors state that “IC and DC sentinels were exposed to donors (1:1) ratio” on line 171, but data presented in Figure 2 indicates one shared donor for both direct and indirect transmission experiments, so use of “1:1 ratio” appears incorrect or misleading here.

4. A clearer depiction of duration of OST treatment needed in figure images/legends. Methods state that OST was delivered twice daily but it is unclear in panel A from figures 3 and 4 if OST treatment was ceased mid-contact with the donor ferret or not. It would be extremely helpful to the reader for the figure legend to include the total number of doses of OST doses received by inoculated ferrets.

5. The discussion section is very informative and well-referenced with regard to the implications of these findings for humans, but it would still be of benefit for laboratorians to see this data presented in the context of other experiments conducted in ferrets. It would be helpful to include (even in just a limited capacity in places like lines 452-60 or 480-83) discussion of how these results compare to other in vivo challenge studies that have examined the ability of antiviral drugs to limit virus transmission to contact ferrets, so readers can understand the effect of BXA treatment relative to what has been shown previously (or not) in the literature with other intervention strategies.

6. The authors discuss on lines 490-1 differences in DC transmission efficiency between the London and Melbourne experiments and provide several potential explanations for this. It would be beneficial to include in the methods section the frequency of bedding change-outs in the two laboratory setups so the reader could better understand the potential for transmission in this system driven by virus-containing fomites in the cages.

7. Lines 155 and 160 in the methods are redundant regarding daily collection of body weights; one of these can be removed.

8. Please specify in the methods volume and gauge needle employed to deliver BXA subcutaneously to ferrets.

9. It is unclear for the ‘placebo’ group in Figures 3-5 how this group was treated; did these ferrets receive sham sub-cutaneous injection (controlling for BXA delivery), sham sugar suspension orally (controlling for OST delivery), or both? This information (and how placebo differs from ‘control’ presented in Figure 2) should be more clearly articulated in the methods. Both ‘control’ and ‘placebo’ ferrets are described as ‘untreated’ in the figure legends so improved detail is needed.

Reviewer #2: 1. Suggested change to title: “Baloxavir treatment of ferrets infected with influenza A(H1N1)pdm09 virus reduces onward transmission”.

2. Line 38 – word “sentinel” is not commonly used in the description of such kind of studies.

3. Materials and methods – Include paragraph entitled “Compounds” and list all antiviral drugs used in the study, their source and method of preparation for PK and antiviral experiments.

4. Line 111 – Is it months or weeks? Please check.

5. Line 118 - Spell out abbreviations LC-MS.

6. Line 138 – Here and throughout the text of the manuscript, change into “A/England/195/2009 (H1N1)pdm09”.

7. Line 154 - provide reference for virus neutralization assay. Whether antibodies to other subtypes and influenza B virus were determined?

8. Line 167 - Indicated the distance between donors and IC animals.

9. Line 180 – Indicated erythrocytes used for HI assays.

10. Lines 210-216 - Indicate that a single dose of BXA was used, and twice daily OST for 5 days were used.

11. Line 303 – Explain why 6.85 ng/mL is the estimated target of effectiveness.

12. Supplementary Table S1 – Indicate in the notes to the Table the meaning “+” and “-“.

Reviewer #3: The experimental details for the ferret model are not completely described for all research sites. The PK studies were conducted at one site, but efficacy studies were conducted at two research institutions. The oral gavage method for delivering Oseltamivir should be described in greater detail.

PLOS authors have the option to publish the peer review history of their article (what does this mean?). If published, this will include your full peer review and any attached files.

Reviewer #1: No

Reviewer #2: No

Reviewer #3: No

---

## [Editor Report · Decision Letter 1]

10 Feb 2020

Dear Dr. Barclay,

Thank you for your thorough responses to the reviewers' comments. We are pleased to inform you that your manuscript 'Baloxavir treatment of ferrets infected with influenza A(H1N1)pdm09 virus reduces onward transmission' has been provisionally accepted for publication in PLOS Pathogens.

Before your manuscript can be formally accepted you will need to complete some formatting changes, which you will receive in a follow up email. A member of our team will be in touch within two working days with a set of requests.

Best regards,

Anice C. Lowen

Associate Editor

PLOS Pathogens

Ron Fouchier

Section Editor

PLOS Pathogens

Kasturi Haldar

Editor-in-Chief

PLOS Pathogens

orcid.org/0000-0001-5065-158X

Michael Malim

Editor-in-Chief

PLOS Pathogens

orcid.org/0000-0002-7699-2064
---

## [Editor Report · Acceptance letter]

9 Mar 2020

Dear Dr. Barclay,

We are delighted to inform you that your manuscript, "Baloxavir treatment of ferrets infected with influenza A(H1N1)pdm09 virus reduces onward transmission," has been formally accepted for publication in PLOS Pathogens.

Best regards,

Kasturi Haldar

Editor-in-Chief

PLOS Pathogens

orcid.org/0000-0001-5065-158X

Michael Malim

Editor-in-Chief

PLOS Pathogens

orcid.org/0000-0002-7699-2064